# New Polyketide and Butenolide Derivatives from the Mangrove Fungus *Aspergillus spelaeus* SCSIO 41433

**DOI:** 10.3390/md23060251

**Published:** 2025-06-13

**Authors:** Zimin Xiao, Jiaqi Liang, Chun Yang, Jian Cai, Bin Yang, Xuefeng Zhou, Jie Yuan, Huaming Tao

**Affiliations:** 1School of Traditional Chinese Medicine, Southern Medical University, Guangzhou 510515, China; 15917491112@163.com (Z.X.); 3201008013@smu.edu.cn (C.Y.); 2Guangdong Key Laboratory of Marine Materia Medica/State Key Laboratory of Tropical Oceanography, South China Sea Institute of Oceanology, Chinese Academy of Sciences, Guangzhou 510301, China; liangjiaqi22@mails.ucas.ac.cn (J.L.); caijian@scsio.ac.cn (J.C.); yangbin@scsio.ac.cn (B.Y.); xfzhou@scsio.ac.cn (X.Z.); 3University of Chinese Academy of Sciences, Beijing 100049, China; 4Zhongshan School of Medicine, Sun Yat-sen University, Guangzhou 510080, China

**Keywords:** mangrove sediment-derived fungi, butenolides, alkaloids, PDE4 inhibitors, anticancer

## Abstract

Two new racemic mixtures, including a polyketide, (±)-penilactone F (**1**), and a butenolide, (±) phenylbutyrolactone IIa (**2**), were isolated from the mangrove sediment-derived strain *Aspergillus spelaeus* SCSIO 41433. Additionally, 20 known compounds were isolated, including four penicillin-like compounds (**11**–**14**), three alkaloids (**15**–**17**), one sesquiterpene (**18**), and four phenolic acids (**19**–**22**). Their structures were elucidated through NMR spectroscopy, HRESIMS, X-ray diffraction, and ECD calculations. In the PDE4 inhibitory activity and anticancer cell activity assays, compounds **2**, **3**, **5**, **8**, **9**, **11**–**14**, and **16** exhibited weak PDE4 inhibitory activity at a concentration of 10 µM, Compound **11** demonstrated potent inhibitory effects against six cancer cell lines (MDA-MB-231, MDA-MB-435, HCT116, SNB-19, PC3, and A549), with IC_50_ values ranging from 3.4 to 23.7 µM.

## 1. Introduction

The ocean represents a vast reservoir of natural resources, and the search for novel natural products from marine environments has become a global research focus in drug discovery. According to recent statistics, a total of 1425 new compounds exhibiting diverse biological activities were identified from marine organisms in 2021 alone [1]. Among these sources, mangrove sediment-derived microorganisms (MSMs) have emerged as a particularly prolific group for the discovery of structurally diverse and bioactive secondary metabolites. By the end of 2021, a total of 519 new natural products had been isolated and characterized from MSMs, of which approximately 57% were fungal-derived. These fungal metabolites exhibited a wide array of potent bioactivities, including antibacterial properties against various pathogens, significant anticancer potential, anti-inflammatory effects, antioxidant activity, as well as antiviral, antiparasitic, and immunomodulatory activities [2].

Mangrove ecosystems, characterized by their dynamic and complex ecological structure, provide a unique and underexplored habitat for microbial communities. These microbial populations play essential roles in the biogeochemical cycling of nutrients within mangrove sediments. The discovery of 519 novel natural products from MSMs, especially those derived from marine fungi, highlighted the immense potential of the marine ecosystem as a rich source of pharmacologically relevant secondary metabolites. The uniqueness of the mangrove environment—defined by high salinity, low oxygen availability, tidal gradients, elevated temperatures, and intense sunlight exposure—has been considered a major driving force behind the metabolic versatility and adaptability of its resident microorganisms [3]. These environmental stressors are believed to stimulate the biosynthesis of a broad spectrum of secondary metabolites with unique chemical scaffolds and potent bioactivities.

Consequently, MSMs have garnered significant attention as a promising and sustainable resource for the discovery of novel bioactive compounds. Continued exploration of the chemical diversity encoded in these microorganisms not only expands the repertoire of marine natural products but also provides valuable leads for the development of new therapeutic agents. Given the increasing incidence of drug-resistant pathogens and the ongoing need for effective anti-inflammatory, anticancer, and antiviral agents, the investigation of secondary metabolites from MSMs remains a highly productive and strategically important area of natural product research.

Pulmonary inflammation is a natural response of the body to injury, aimed at eliminating harmful stimuli such as pathogens, irritants, and damaged cells, while initiating the healing process. Both acute and chronic pulmonary inflammation are observed in various inflammatory airway diseases, such as acute respiratory distress syndrome, chronic obstructive pulmonary disease (COPD), asthma, and cystic fibrosis (CF), all of which severely impact human health [4]. Since the early 1980s, phosphodiesterase 4 (PDE4) has been an attractive target for the treatment of inflammatory airway diseases. Researchers have found that PDE4 inhibitors exhibit broad-spectrum anti-inflammatory activity, along with additional bronchodilatory and neuroregulatory effects, showing significant efficacy in treating inflammatory airway diseases [5]. However, PDE4 inhibitors are not without drawbacks, as their severe side effects have been a major challenge for researchers to overcome.

In recent years, new compounds isolated from marine fungi, such as sanshamycins A–E and talaroterphenyls A−D, have been shown to possess PDE4 inhibitory activity, and they are promising candidates for the development of new PDE4 inhibitors with fewer side effects [6,7,8]. Clearly, searching for new PDE4 inhibitors with minimal side effects from secondary metabolites of marine fungi offers an effective solution to address the current problem of severe side effects associated with PDE4 inhibitors.

*Aspergillus* species are among the more common fungi found in marine environments, exhibiting a broad range of species and potential biological activities. Their secondary metabolites, characterized by unique chemical structures and novel bioactivities, have attracted significant attention and are considered an important source for drug discovery. In this study, 22 compounds were isolated and identified from the rice fermentation extract of the mangrove sediment-derived *Aspergillus spelaeus* SCSIO 41433. Among these, 13 polyketides (**1**–**10**) were identified, including a polyketide, (±)-penilactone F (**1**), and a butenolide, (±) phenylbutyrolactone IIa (**2**). Additionally, four penicillin-like compounds (**11**–**14**), three alkaloids (**15**–**17**), one sesquiterpene (**18**), and four phenolic acids (**19**–**22**) were also isolated (Figure 1).

## 2. Results

### 2.1. Structural Elucidation

Compound **1** was colorless oil. The HR-ESIMS ion peak at *m*/*z* 247.0943 [M + Na]⁺ (calculated for C_12_H_16_O_4_Na, 247.0941) suggested a molecular formula of C_12_H_16_O_4_ with five degrees of unsaturation. The ^1^H NMR spectra (Table 1) showed signals for four aromatic protons at *δ* 7.08 (d, *J* = 8.1 Hz, H-4, 8) and *δ* 6.72 (d, *J* = 8.0 Hz, H-5, 7), two secondary methylene protons at *δ* 4.87 (m, H-9) and *δ* 3.85 (qd, H-10), a methylene proton at *δ* 3.54 (s, H-2), and two methyl protons at *δ* 1.18 (H_3_, d, *J* = 6.5 Hz, H-11) and *δ* 1.11 (H_3_, d, *J* = 6.5 Hz, H-12). The ^13^C NMR and HSQC spectra (Table 1) showed 12 carbon signals, including an ester carbonyl at *δ* 172.4 (C-1), six aromatic carbons at *δ* 155.3 (C-6), 130.4 (C-4, C-8), 125.6 (C-3), and 115.7 (C-5, C-7), two oxygenated secondary methylene carbons at *δ* 74.9 (C-9) and 69.7 (C-10), a methylene carbon at *δ* 40.9 (C-2), and two methyl carbons at *δ* 17.6 (C-12) and 14.2 (C-11).

The ^1^H-^1^H COSY correlations observed between *δ* 7.08 (H-4, H-8) and *δ* 6.72 (H-5, H-7) suggested the presence of a disubstituted aromatic ring. The HMBC correlations from H-2 to C-1 and from H-11, H-12 to C-9 and C-10 (Figure 2), along with the ^1^H-^1^H COSY analysis of H-11/H-9/H-10/H-12, led to the elucidation of the side chain’s planar structure. Additionally, HMBC correlations from H-2 to C-3, C-4, and C-8 confirmed the connection between the side chain and the disubstituted aromatic ring. Therefore, the planar structure of compound **1** was established, which corresponds to the previously reported structure of penilactone D [9].

Compound **1** has two stereocenter carbons (C-9 and C-10). However, the optical rotation of the mixture was close to zero, and the lack of a significant Cotton effect in the electronic circular dichroism (ECD) spectrum suggests that these compounds are likely a pair of enantiomers. To confirm the stereochemistry of compound **1**, the chemical shift difference (Δ*δ*c = 2.8) between the two methyl groups in the syn isomer was smaller than that in the anti isomer (Δ*δ*c = 3.6), as reported in the literature [9]. The ^13^C NMR data showed a shift difference of Δ*δ*c = 3.4, indicating that the compounds are the anti isomers, where the two methyl groups are oriented in the same direction. Therefore, the relative configurations were determined to be rel-(9*R*, 10*S*) for (+)−**1** and rel-(9*S*, 10*R*) for (−)−**1**. Finally, based on ECD calculations (Figure 3), the absolute configuration of (+)−**1** was determined to be 9*R*, 10*S*, and that of (−)−**1** was 9*S*, 10*R*. (+)−**1** was named (+)−penilactone F, and (−)−**1** was named (−)−penilactone F.

Compound **2** was obtained as a white powder. High-resolution electrospray ionization mass spectrometry (HR-ESIMS) revealed a deprotonated molecule peak at *m*/*z* 309.0760 [M-H]^-^ (calculated for C_18_H_13_O_5_, 309.0768), suggesting a molecular formula of C_18_H_14_O_5_ with 12 degrees of unsaturation. The ^1^H NMR data (Table 1) showed a methylene proton signal at *δ*_H_ 3.56 (s, H_2_-6) and ten protons from five pairs of aromatic protons at *δ*_H_ 7.78 (m, H-14, 18), 7.46 (t, *J* = 7.8 Hz, H-15, 17), 7.39 (m, H-10), 7.12 (m, H-16), 7.09 (dd, *J* = 8.2, 6.5 Hz, H-9, 11), and 6.83 (m, H-8, 12). These proton signals suggested the presence of two isolated phenyl rings.

Analysis of the ^13^C, DEPT-135 and HSQC spectra revealed 18 carbon signals, including two carbonyl carbons at *δ*_C_ 172.4 (C-1) and 170.1 (C-5), two olefinic carbons at *δ*_C_ 128.8 (C-3) and 141.8 (C-4) (including one oxidized carbon), a methylene carbon at *δ*_C_ 40.1 (C-6), an oxygenated quaternary carbon at *δ*_C_ 87.0 (C-2), and twelve aromatic carbons at *δ*_C_ 135.1 (C-7), 132.0 (C-13), 131.5 (C-8, C-12), 129.7 (C-10, C-15, C-17), 128.8 (C-9, C-11, C-14, C-18), and 128.0 (C-16).The structure was further deduced by the HMBC correlations from H-6 to C-1, C-2, C-7, C-8, and C-12, indicating that the substituents at C-2 include an ethylphenyl group and a carboxyl group. The HMBC correlations from H-14, 18 to C-3 suggest a phenyl group attached to C-3. Based on this data, the core structure of the compound was proposed as a butenolide, with a hydroxyl group attached at C-4. Thus, the planar structure of compound **2** was deduced (Figure 2).

Despite having a stereocenter at C-2, the optical rotation of compound **2** was close to zero, and there was no significant Cotton effect in the ECD spectrum, suggesting that these may be a pair of enantiomers. The results confirmed that compound **2** are enantiomers with a 1:1 ratio (Figure 4). The “*S*” configuration corresponds to compound **2** named phenylbutyrolactone IIa, which has been reported in biosynthesis studies [10], while the “*R*” configuration is a new discovery. The chemical structure of compound **2** was further confirmed by X-ray crystallography (Figure 4). Unfortunately, due to the low separation efficiency of the chiral column, enantiomers of compound **2** could not be isolated individually.

By comparing their physicochemical properties and spectroscopic data with the reported literature values, other known compounds were determined. Compounds present in SCSIO 41,433 were asperteretal G (**3**) [11], nafuredin (**4**) [12], methyl 2-(3,4-dihydroxyphenyl)acetate (**5**) [13], methyl 2-(4-hydroxyphenyl)acetate (**6**) [14], methyl 2-(1*H*-indol-3-yl)acetate (**7**) [15], 5-hydroxy-7-methoxy-2,3-dimethyl-4*H*-chromen-4-one (**8**) [16], 5-hydroxy-3-(hydroxymethyl)-7-methoxy-2-methyl-4*H*-chromen-4-one (**9**) [16], 4-hydroxy-3,6-dimethyl-2*H*-pyran-2-one (**10**) [17], trichodermamide B (**11**) [18], aspergillazine A (**12**) [19], trichodermamide D (**13**) [20], trichodermamide A (**14**) [18], *N*-(2-(1*H*-indol-3-yl)ethyl)acetamide (**15**) [21], 1*H*-indole-3-carbaldehyde (**16**) [22], 1*H*-pyrrole-2-carboxylic acid (**17**) [23], (1*R*,2*R*,3*S*,4*R*)-1,2,3-trimethyl-4-(4-methylpent-3-en-1-yl)cyclohexane-1,3-diol (**18**) [24], (E)-3-(4-hydroxy-3-methoxyphenyl)acrylic acid (**19**) [25], 2-phenylacetic acid (**20**) [26], 2-(4-methoxyphenyl)acetic acid (**21**) [27], and 2-(4-hydroxyphenyl)acetic acid (**22**) [28]; related physicochemical and spectroscopic data is shown in Appendix A.

### 2.2. Biological Activity Assay

Cyclic adenosine monophosphate (cAMP) is an important signaling molecule involved in various cellular processes, including inflammation, immune responses, and smooth muscle contraction. Phosphodiesterase 4 (PDE4) plays a key role in regulating cAMP levels and is considered an important drug target [29]. In this study, we preliminarily evaluated the PDE4 inhibitory activity of compounds **2**, **3**, **5**, **8**, **9**, **11**–**14**, and **16** at a concentration of 10 μM using the PDE4 scintillation proximity assay (SPA) (Table 2).

Additionally, compounds **1**–**4**, **7**–**9**, **11**, and **15**–**21** were screened for anticancer activity in vitro, with cisplatin (cis-platinum) as a positive control. The results showed that compound **11** exhibited strong anticancer activity to human breast cancer cells (MDA-MB-231, MDA-MB-435), human colon cancer cells (HCT116), human glioma cells (SNB-19), human prostate cancer cells (PC3), and human non-small cell lung cancer cells (A549) (Table 3); among them, it exhibited a particularly strong inhibitory effect on SNB-19, with an IC_50_ of 3.4 ± 0.1.

## 3. Materials and Methods

### 3.1. General Experimental Procedures

UV spectra were recorded using an 8453VU-Vis UV-Visible Spectrophotometer (Agilent, Beijing, China). IR spectra were obtained using an IR Affinity-1 Spectrometer (Shimadzu, Beijing, China). HRESIMS spectra were recorded on a Bruker maXis Q-TOF Mass Spectrometer (Bruker, Fallanden, Switzerland). NMR spectra were recorded on an AVANCE III HD 600 MHz spectrometer (Bruker BioSpin International AG, Fällanden, Switzerland), with chemical shifts expressed in δ values. Optical rotations were determined using an Anton Paar MPC 500 polarimeter (Hertford, UK). HPLC was performed on an Agilent 1260 system equipped with a DAD detector, using an ODS column (YMC pack ODS-a, 10 × 250 mm, 5 µm). Thin-layer chromatography (TLC) and column chromatography (CC) were performed on plates pre-coated with silica gel GF254 (10–40 µm), silica gel (200–300 mesh) (Qingdao Marine Chemical Factory, Qingdao, China), and Sephadex LH-20 (Amersham Biosciences, Uppsala, Sweden). Spots were detected under UV light at 254 nm using TLC plates from Qingdao Marine Chemical Factory. All solvents used, except for the mobile phase of HPLC, were of analytical grade (Tianjin Fuyu Chemical Factory, Tianjin, China). HPLC-grade solvents (Shanghai Xingke High-Purity Solvent Co., Ltd., Shanghai, China) were used for HPLC. X-ray diffraction was performed on an XtalLAB PRO diffractometer (Rigaku Corporation, Tokyo, Japan) with Cu Kα radiation.

### 3.2. Fungal Source and Strain Identification

The fungal strain SCSIO 41433 was isolated from the Gaoqiao Mangrove Wetland (21.573° N, 109.767° E) located along the northern coastline of the Beibu Gulf, Zhanjiang, China. The strain was stored on MB agar slants (malt extract 15 g, sea salt 10 g, agar 16 g, H_2_O 1 L, pH 7.4–7.8) in liquefied mineral oil, and was preserved at the Key Laboratory of Tropical Marine Biological Resources and Ecology, South China Sea Institute of Oceanology, Chinese Academy of Sciences. The strain SCSIO 41433 was identified as *Aspergillus spelaeus* based on its ITS sequence (GenBank accession number PP407040), which shares 100% similarity with *Aspergillus spelaeus* MG976683.1.

### 3.3. Fungal Cultivation and Fermentation

The fermentation of the *Aspergillus spelaeus* SCSIO 41433 strain was carried out using solid culture media. The preparation method for the solid culture medium was as follows: 180 mL distilled water, 3 g sea salt, and 150 g rice were added to a 1000 mL Erlenmeyer flask. For the MB seed liquid, the preparation method was as follows: 400 mL distilled water, 8 g sea salt, and 6 g malt powder were added to a 1000 mL Erlenmeyer flask, and the pH was adjusted to 7.4–7.8. Both media were sterilized at 121 °C for 30 min.

The strain fermentation method involved inoculating the preserved strain onto MB agar plates to activate it. After culturing at 26 °C for 5 days, agar pieces (1–2 cm^2^) with newly grown *Aspergillus spelaeus* SCSIO 41433 were selected from the MB agar plates and transferred to the MB seed liquid for cultivation. Two flasks of seed liquid were prepared and incubated on a shaking incubator at 27 °C, 180 rpm, for 48 h. The seed liquid was then transferred to rice solid culture media and incubated at 26 °C for 30 days.

### 3.4. Extraction and Separation

The fermentation products were crushed, extracted with ethyl acetate by ultrasonic treatment, and vacuum concentrated to yield a crude extract (130.0 g). Medium pressure liquid chromatography (MPLC) with silica gel (200–300 mesh) was used for column chromatography, employing a gradient of petroleum ether/dichloromethane (0–100%, *v*/*v*) and dichloromethane/methanol (0–100%, *v*/*v*). Nine fractions (Frs. A–I) were obtained, as confirmed by TLC.

Fr. B (8.2 g) was subjected to Sephadex LH-20 column chromatography with methanol as the eluent, yielding three sub-fractions (Frs. B1–3). Fr. B3 was further purified by semi-preparative HPLC (CH_3_CN/H_2_O, 58/42) to obtain **8** (6.8 mg, *t*_R_ = 16.8 min).

Fr. C (5.5 g) was separated by ODS CC using a gradient of CH_3_OH/H_2_O (15/85–100/0), yielding three sub-fractions (Frs. C1–3). Fr. C1 was further purified by semi-preparative HPLC (CH_3_OH/H_2_O, 66/34) to obtain **7** (1.3 mg, *t*_R_ = 24.5 min).

Fr. D (7.1 g) was processed using Sephadex LH-20 column chromatography with methanol/dichloromethane (1/1) as the eluent, yielding five sub-fractions (Frs. D1–5). Fr. D1 was purified by semi-preparative HPLC (CH_3_OH/H_2_O, 83/17, 0.1% HCOOH) to obtain **4** (5.7 mg, *t*_R_ = 21.3 min). Frs. D2–5 (0.9 g) were separated by ODS CC (CH_3_OH/H_2_O, 20/80–100/0) to yield three sub-fractions (Frs. D2.1–D2.3). Fr. D2.2 was further purified by semi-preparative HPLC (CH_3_OH/H_2_O, 33/67, 0.1% HCOOH), yielding **21** (4.2 mg, *t*_R_ = 25.0 min), **16** (3.4 mg, *t*_R_ = 28.2 min), and **20** (8.0 mg, *t*_R_ = 31.9 min). Fr. D2.3 was purified with CH_3_CN/H_2_O (33/67, 0.1% HCOOH), yielding **9** (5.9 mg, *t*_R_ = 17.3 min) and **3** (11.3 mg, *t*_R_ = 26.3 min), identified as a pair of enantiomers by chiral HPLC.

Fr. E (1.7 g) was separated by Sephadex LH-20 column chromatography with methanol as the eluent, yielding four sub-fractions (Frs. E1–4). Fr. E1 was further separated using silica gel column chromatography (petroleum ether/ethyl acetate, 10/1) to yield **18** (8.8 mg). Fr. E3 was separated by semi-preparative HPLC (CH_3_OH/H_2_O, 33/67, 0.1% HCOOH), yielding two sub-fractions (Frs. E3.1–E3.2). Fr. E3.1 was purified with CH_3_OH/H_2_O (20/80) to yield **17** (9.5 mg, *t*_R_ = 18.1 min), and Fr. E3.2 with CH_3_OH/H_2_O (60/40, 0.1% HCOOH) yielded **2** (7.4 mg, *t*_R_ = 23.5 min). Fr. E4 was purified by HPLC (CH_3_OH/H_2_O, 35/65, 0.1% HCOOH), yielding **19** (5.6 mg, *t*_R_ = 23.1 min).

Fr. F (4.0 g) was subjected to Sephadex LH-20 column chromatography with methanol, yielding three sub-fractions (Frs. F1–3). Fr. F2 was purified by HPLC (CH_3_OH/H_2_O, 45/55, 0.1% HCOOH), yielding **15** (0.9 mg, *t*_R_ = 14.8 min).

Fr. G (6.0 g) was separated by Sephadex LH-20 column chromatography with methanol, yielding three sub-fractions (Frs. G1–3). Fr. G1 was further purified by semi-preparative HPLC (CH_3_OH/H_2_O, 38/62) to obtain **1** (22.0 mg, *t*_R_ = 23.6 min) and **10** crude product, which was purified to yield **10** (1.2 mg, *t*_R_ = 18.7 min). Part of **1** was separated using a chiral column (CHIRALPAK IC, 4.6 mmI.D.×250 mmL, 5 µm) with isopropanol/n-hexane (10/90) as the mobile phase to obtain (+)−**1** (2.1 mg) and (−)−**1** (1.8 mg). Fr. G2 was purified by HPLC (CH_3_OH/H_2_O, 28/72, 0.1% HCOOH), yielding **22** (20.9 mg, *t*_R_ = 16.2 min), and Fr. G3 was purified by HPLC (CH_3_CN/H_2_O, 33/67, 0.1% HCOOH), yielding **11** (12.0 mg, *t*_R_ = 32.9 min).

Fr. H (4.5 g) was separated by Sephadex LH-20 column chromatography with methanol, yielding three sub-fractions (Frs. H1–3). Fr. H1 was separated by semi-preparative HPLC (CH_3_OH/H_2_O, 35/65) to obtain **5** (1.3 mg, *t*_R_ = 13.7 min) and **6** (3.5 mg, *t*_R_ = 22.5 min). Fr. H2 was purified by HPLC (CH_3_CN/H_2_O, 25/75, 0.1% HCOOH), yielding **13** (6.1 mg, *t*_R_ = 23.4 min), and Fr. H3 was purified with CH_3_OH/H_2_O (47/53, 0.1% HCOOH) to obtain **12** (4.9 mg, *t*_R_ = 15.0 min).

Fr. I (2.6 g) was separated by Sephadex LH-20 column chromatography with methanol, yielding three sub-fractions (Frs. I1–3). Fr. I2 was further purified by semi-preparative HPLC (CH_3_OH/H_2_O, 43/57), yielding **14** (7.0 mg, *t*_R_ = 18.3 min).

### 3.5. Physicochemical Data of New Compounds

Penilactone F (**1**): colorless oil; (+)−**1**
${\left[\alpha \right]}_{D}^{25}$ + 0.9 (*c* 0.1, CH_3_OH); ECD (0.2 mg/mL, CH_3_OH) *λ*_max_ (∆*ε*) 237 (−0.93); (−)−**1**
${\left[\alpha \right]}_{D}^{25}$ − 2.0 (*c* 0.1, CH_3_OH); ECD (0.2 mg/mL, CH_3_OH) *λ*_max_ (∆*ε*) 229 (0.76); UV (CH_3_OH) λ_max_ (log *ε*) 225 (1.29) nm; IR*υ*_max_: 3335, 2980, 2941, 2833, 1715, 1597, 1516, 1227, 1022, 802, 694 cm^−1^; ^1^H and ^13^C NMR data as shown in Table 1; HRESIMS *m*/*z* 247.0943 [M+Na]^+^ (calculated for C_12_H_16_O_4_Na, 247.0941).

Phenylbutyrolactone IIa (**2**): white powder; ${\left[\alpha \right]}_{D}^{25}$ + 0.8 (*c* 0.1, CH_3_OH); UV (CH_3_OH) λ_max_ (log *ε*) 288 (0.41) nm; IR*υ*_max_: 3375, 3064, 3032, 2928, 1730, 1695, 1603, 1024, 712, 696 cm^−1^; ^1^H and ^13^C NMR data as shown in Table 1; HRESIMS *m*/*z* 309.0760 [M-H]^-^ (calculated for C_18_H_13_O_5_, 309.0768). Crystal data for C_18_H_14_O_5_ (*M* = 310.29 g/mol): tetragonal, space group I4_1_/a (no. 88), *a* = 27.4851(3) Å, *c* = 7.72760(10) Å, *V* = 5837.67(15) Å^3^, *Z* = 16, *T* = 99.98(18) K, μ(Cu Kα) = 0.863 mm^−1^, *D*_calc_ = 1.412 g/cm^3^, 21,376 reflections measured (6.432° ≤ 2Θ ≤ 148.668°), 2950 unique (*R*_int_ = 0.0357, *R*_sigma_ = 0.0202) which were used in all calculations. The final *R*_1_ was 0.0451 (*I* > 2*σ*(*I*)) and *wR*_2_ was 0.1080 (Figure 4, CCDC 2449655).

### 3.6. ECD Calculation

The conformational analysis of compound **1** was performed using Spartan’14 software (v1.14, Wavefunction, Irvine, CA, USA) with the Molecular Merck force field. Subsequently, Gaussian09 (D.01, Pittsburgh, PA, USA) was used to optimize the conformations with a Boltzmann population greater than 1% in methanol using the PCM model [9]. At the B3LYP/6-31G(d) level, the stable conformers were further optimized. ECD calculations were then performed at the B3LYP/6-311G (d,p) level for the optimized stable conformations. The rotational intensities for a total of 20 excited states were calculated. The overall ECD data was weighted by the Boltzmann distribution, and the contribution of each conformer’s isomer was based on the UV-corrected Boltzmann computation. The ECD curves and enantiomeric ECD curves were generated using GaussView 6.0 software with a half-width of 0.33 eV [30].

### 3.7. PDE4 Inhibition Rate Assay

As described previously, PDE4 activity was evaluated using a PDE4 scintillation proximity assay [31]. Briefly, 3H-cAMP or 3H-cGMP was used as the substrate to assess enzyme activity of the catalytic domain. The assay buffer contained Tris-HCl, MgCl_2_ or MnCl_2_, DTT, and 3H-cAMP or 3H-cGMP at a protein concentration of 2 nM. After a 15-min incubation, the reaction was terminated, and the reaction products were precipitated, with unreacted substrates remaining in the supernatant. The radioactivity of the supernatant was measured using a liquid scintillation counter, and the inhibition rate of the compound at 10 µM concentration was calculated.

### 3.8. Cell Culture

Human cancer cell lines MDA-MB-231, MDA-MB-435, HCT116, SNB-19, A549 and PC3 were procured from Shanghai Institute of Biological Sciences (Shanghai, China). MDA-MB-231, MDA-MB-435, HCT116, SNB-19 and A549 were maintained in Dulbecco’s Modified Eagle Medium (DMEM) (Invitrogen, Carlsbad, CA, USA). PC3 was cultured in Ham’s F-12K (Invitrogen, Carlsbad, CA, USA). All cell culture media were supplemented with 10% fetal bovine serum (Hyclone, Logan, UT, USA), 2 mM L-glutamine, 100 mg/mL streptomycin and 100 units/mL penicillin (Invitrogen, Carlsbad, CA, USA). The cultures were maintained at 37 °C in a humidified atmosphere of 5% CO_2_.

### 3.9. Antitumor Experiment

The cytotoxicity of some compounds against human breast cancer cells (MDA-MB-231, MDA-MB-435), human colon cancer cells (HCT116), human glioma cells (SNB-19), human prostate cancer cells (PC3), and human non-small cell lung cancer cells (A549) was evaluated using the MTT assay to assess cell viability. Briefly, cells were seeded in 96-well plates at a density of 5 × 10^3^ cells per well and incubated overnight, followed by treatment with the compounds for the required duration (24 h). The optical density (OD 570) at 570 nm was measured using the Hybrid Multi-Mode Reader (Synergy H1, BioTek, Santa Clara, CA, USA). The experiment was independently repeated three times.

## 4. Conclusions

In summary, microorganisms that have resided long-term in mangrove ecosystems gradually developed unique and complex metabolic pathways to adapt to the distinctive and often extreme environmental conditions characteristic of mangrove habitats. These environmental factors include high salinity, fluctuating oxygen levels, tidal influences, and intense sunlight exposure. As a result of these adaptations, these microorganisms have been able to biosynthesize numerous secondary metabolites possessing novel and diverse chemical structures, many of which exhibit a wide range of biological activities [3]. Natural products derived from microorganisms isolated from mangrove soils or sediments have continually demonstrated broad potential applications in pharmaceutical and biotechnological fields. In particular, MSMs have served as a highly effective and valuable source for producing bioactive compounds with unprecedented carbon skeletons, thus playing a critical role in the identification and development of lead compounds for new drug discoveries and therapeutic innovations.

In this study, 22 compounds were isolated from the mangrove sediment fungus *Aspergillus spelaeus* SCSIO 41433, including three new compounds, (±)-penilactone F (**1**), and (±) phenylbutyrolactone IIa (**2**). The absolute configurations of the new compounds were confirmed through ECD calculations, NMR data analysis, and X-ray crystallography. Some compounds were screened for PDE4 inhibitory activity and antitumor activity in vitro. It was found that compound **11** showed substantial antitumor activity, and compounds **2**, **3**, **5**, **8**, **9**, **11**–**14**, and **16** exhibited PDE4 inhibitory activity at a concentration of 10 µM, suggesting their potential as PDE4 inhibitors and warranting further research.

## Figures and Tables

**Figure 1 marinedrugs-23-00251-f001:**
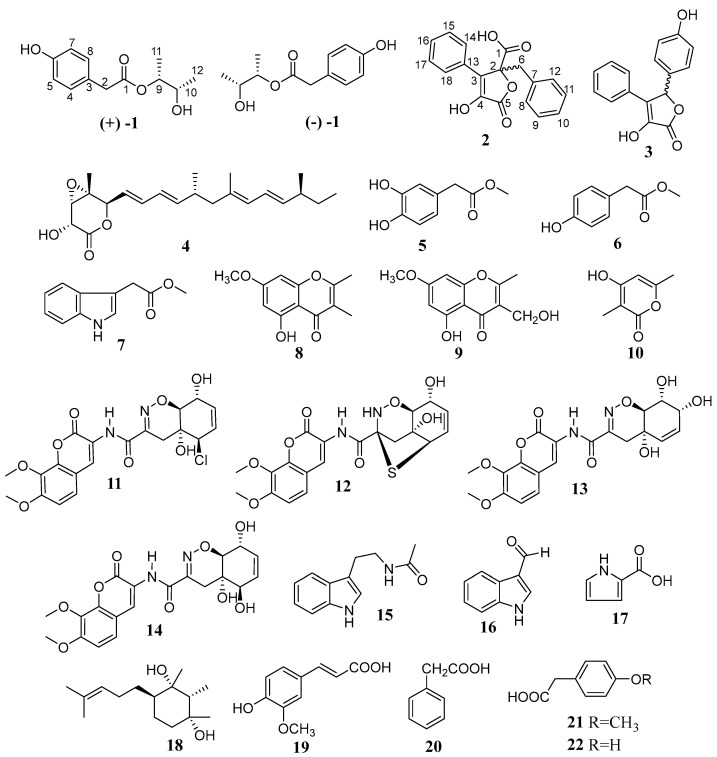
Structures of compounds **1**–**22**.

**Figure 2 marinedrugs-23-00251-f002:**
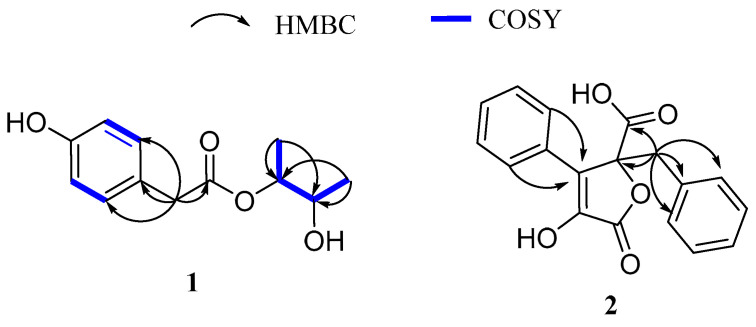
Key ^1^H-^1^H COSY and HMBC correlations of **1** and **2**.

**Figure 3 marinedrugs-23-00251-f003:**
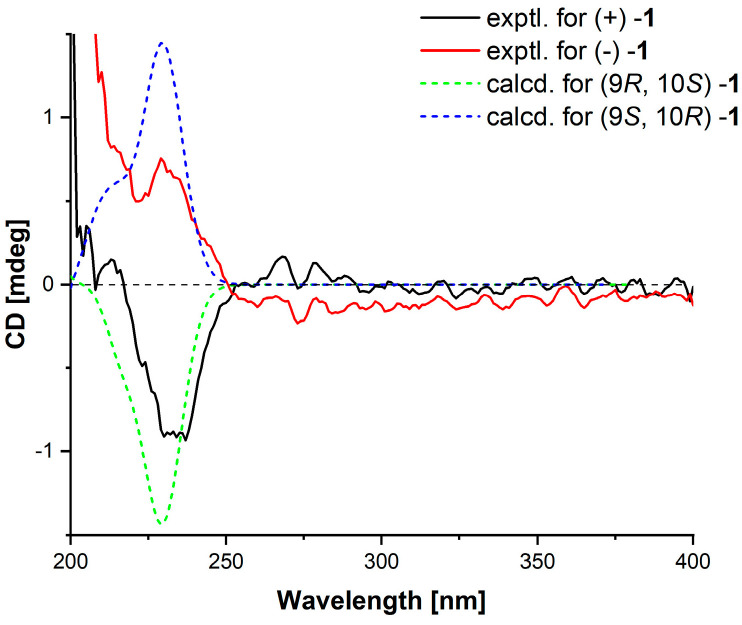
Experimental ECD spectra and calculational ECD spectrum of **1**.

**Figure 4 marinedrugs-23-00251-f004:**
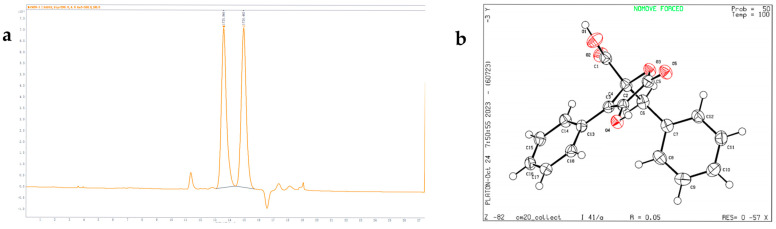
HPLC analysis (**a**) and X-ray crystal diffraction (**b**) of **2**.

**Table 1 marinedrugs-23-00251-t001:** The NMR data of **1** (CDCl_3_) and **2** (CD_3_OD) (600 and 150 MHz, *δ* in ppm).

Pos.	1		2	
	*δ*_C_, Type	*δ*_H_, (*J* in Hz)	*δ*_C_, Type	*δ*_H_, (*J* in Hz)
1	172.4, C		172.4, C	
2	40.9, CH_2_	3.54, s	87.0, C	
3	125.6, C		128.8, C	
4	130.4, CH	7.08, d (8.1)	141.8, C	
5	115.7, CH	6.72, d (8.0)	170.1, C	
6	155.3, C		40.1, CH_2_	3.56, s
7	115.7, CH	6.72, d (8.0)	135.1, CH	
8	130.4, CH	7.08, d (8.1)	131.5, CH	6.83, m
9	74.9, CH	4.87, m	128.8, CH	7.09, dd (8.2, 6.5)
10	69.7, CH	3.85, qd (6.5, 3.1)	129.73, CH	7.39, m
11	14.2, CH_3_	1.18, d (6.5)	128.8, CH	7.09, dd (8.2, 6.5)
12	17.6, CH_3_	1.11, d (6.5)	131.5, CH	6.83, m
13			132.0, C	
14			128.8, CH	7.78, m
15			129.7, CH	7.46, t (7.8)
16			128.04, CH	7.12, m
17			129.7, CH	7.46, t (7.8)
18			128.8, CH	7.78, m

**Table 2 marinedrugs-23-00251-t002:** Inhibition rate of PDE4 by active compounds.

Compound	Inhibition Rate (%)	Compound	Inhibition Rate (%)
**2**	14.5	**11**	18.0
**3**	19.4	**12**	12.7
**5**	9.4	**13**	12.7
**8**	9.5	**14**	11.7
**9**	19.2	**16**	19.4

**Table 3 marinedrugs-23-00251-t003:** Cytotoxicity of compounds on multiple tumor cells.

Tested Compounds					
Cells (IC_50_ ± SD, µM)	1–4	7–9	11	15–21	cis-Platinum
MDA-MB-231	/	/	7.7 ± 0.5	/	44.4 ± 3.3
HCT116	/	/	19.1 ± 1.2	/	38.2 ± 5.9
MDA-MB-435	>50	>50	9.2 ± 0.1	>50	14.7 ± 1.3
SNB-19	/	/	3.4 ± 0.1	/	26.4 ± 7.6
PC3	/	/	23.7 ± 0.5	/	26.5 ± 1.5
A549	/	/	5.7 ± 0.6	/	33.0 ± 0.2

## Data Availability

The data presented in this study is available on request from the corresponding author.

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
