# Peer review of "New Polyketide and Butenolide Derivatives from the Mangrove Fungus Aspergillus spelaeus SCSIO 41433"

_marinedrugs, 2025, doi:10.3390/md23060251_

Round 1
Reviewer 1 Report
Comments and Suggestions for Authors
In the manuscript of Z. Hiao, et al., “New Polyketide and Butenolide Derivatives from the Mangrove Fungus Aspergillus spelaeus SCSIO 41433”, a study of Aspergillus spelaeus strain SCSIO 41433 is reported. Two new racemic mixtures, including a polyketide, (±) -penilactone F and a butenolide, (±) phenylbutyrolactone IIa were isolated from the mangrove sediment-derived strain A. spelaeus SCSIO 41433 along with 20 known compounds, including four penicillin-like compounds three alkaloids, one sesquiterpene, and four phenolic acids. The structures of these compounds were elucidated based on NMR and HRMS data. Absolute configurations were determined through quantum chemical ECD calculations and X-ray crystallography. These compounds were tested for PDE4 inhibitory activity and anticancer cell activity. Trichodermamide B exhibited strong anticancer activity to human breast cancer cells (MDA-MB-231, MDA-MB-435), human colon cancer cells (HCT116), human glioma cells (SNB-19), human prostate cancer cells (PC3), and human non-small cell lung cancer cells (A549).
Reviewer’s notes. (unfortunately, the text of the manuscript has no line enumeration).
- Abstract. “The planar structures of these compounds… their absolute configurations…”: if the structure is planar, it cannot be chiral. Maybe, the authors mean something different? Please, reformulate the phrase.
- Page 3, line 2 below Table 1. “Quasi-molecular ion”: according to IUPAC MS terms glossary, it should be “deprotonated molecule”.
- Page 3, lines 7—8 below Table 1. “…presence of a mono-substituted biphenyl structure”: there is no biphenyl fragment in 2; there are two isolated phenyl rings (indeed, one is p-substituted). Please, change this phrase.
- Page 6, Materials and Methods, lines 12 and 14. “…liquid phase mobile phase”: the repeat of word “phase” is excessive.
- Page 7, Materials and Methods, lines 2—3 in 3.4. MPLC means medium pressure liquid chromatography (not “performance”), in fact, the author mean flash chromatography.
- Page 9. Ref. 9, Aspergillus niger; Ref. 11, Aspergillus costaricaensis: should be italicized.

Author Response
Comment 1: Abstract. “The planar structures of these compounds… their absolute configurations…”: if the structure is planar, it cannot be chiral. Maybe, the authors mean something different? Please, reformulate the phrase.
Response 1: Thank you for pointing this out. We agree with this comment. Therefore, we have changed the phrase to “Their structures were elucidated through NMR spectroscopy, HRESIMS, X-ray diffraction, and ECD calculations.”
Comment 2: Page 3, line 2 below Table 1. “Quasi-molecular ion”: according to IUPAC MS terms glossary, it should be “deprotonated molecule”.
Response 2: Thanks for pointing out the mistake. We have modified the term from “Quasi-molecular ion” to “deprotonated molecule”.
Comment 3: Page 3, lines 7—8 below Table 1. “…presence of a mono-substituted biphenyl structure”: there is no biphenyl fragment in 2; there are two isolated phenyl rings (indeed, one is p-substituted). Please, change this phrase.
Response 3: Thanks for pointing out the mistake. We have modified the phrase from “a mono-substituted biphenyl structure” to “two isolated phenyl rings”.
Comment 4: Page 6, Materials and Methods, lines 12 and 14. “…liquid phase mobile phase”: the repeat of word “phase” is excessive.
Response 4: Thank you for pointing this out. We have changed the phrase to “mobile phase of HPLC.”
Comment 5: Page 7, Materials and Methods, lines 2—3 in 3.4. MPLC means medium pressure liquid chromatography (not “performance”), in fact, the author mean flash chromatography.
Response 5: Thanks for pointing out the mistake. We have modified the phrase from “Medium performance liquid chromatography” to “Medium pressure liquid chromatography”.
Comment 6: Page 9. Ref. 9, Aspergillus niger; Ref. 11, Aspergillus costaricaensis: should be italicized.
Response 6: Thanks for pointing out the mistake. We have made the modification.

Reviewer 2 Report
Comments and Suggestions for Authors
The work submitted for review concerns the analysis of compounds produced by the fungus Aspergillus spelaeus occurring in marine sediments near China, in the Beibu Bay.
The work contains a description of a carefully conducted search for substances secreted by the fungus, their identification, and the determination of the full structure of two previously unknown compounds called penilactone F and phenylbutyrolactone IIa.
The Authors used various chromatographic techniques to isolate pure substances (fungal metabolites), which require time and precision.
Having already detected and purified all of them, they were tested as potential anticancer preparations against six selected cancer cell lines.
Only trichodermamide B, known for 22 years, showed clear activity, significantly higher than the reference cis-platin.
It was particularly active against SNB-19 cells (human glioma cells, effective more than 7 times) and A549 (non-small cell lung cancer).
The presented work is done reliably, the procedures used are clearly described.
I found no serious errors in it.
I only have a comment that the use of ultrasound at the extraction stage can lead to the destruction of more sensitive natural compounds, so the image of what is in the environment can be distorted.
I also have one comment regarding the equipment used. In point 3.6 the Authors described the ECD calculations, but in point 3.1 they forgot to write what apparatus they used to measure optical rotation.
I believe that after supplementing this information the article should be published.
Author Response
Comment 1: I only have a comment that the use of ultrasound at the extraction stage can lead to the destruction of more sensitive natural compounds, so the image of what is in the environment can be distorted.
Response 1: Thank you for pointing this out. We agree with this comment. Therefore, We will consider a gentler extraction method.
Comment 2: I also have one comment regarding the equipment used. In point 3.6 the Authors described the ECD calculations, but in point 3.1 they forgot to write what apparatus they used to measure optical rotation.
Response 2: Thanks for pointing out the mistake. We have replenished the relevant apparatus.
